# EEG-Based Prediction of Stress Responses to Naturalistic Decision-Making Stimuli in Police Cadets

**DOI:** 10.3390/s25185925

**Published:** 2025-09-22

**Authors:** Abdulwahab Alasfour, Nasser AlSabah

**Affiliations:** 1Department of Electrical Engineering, College of Engineering and Petroleum, Kuwait University, Kuwait City, Kuwait; 2John Jay College of Criminal Justice, CUNY Graduate Center, New York City, NY 10019, USA; 3Kuwait National Police, Ministry of Interior, Kuwait City, Kuwait

**Keywords:** decision-making, EEG, heart rate variability, neurophysiology, stress, biomedical signal processing

## Abstract

The ability of police officers to make correct decisions under emotional stress is critical, as errors in high-pressure situations can have severe legal and physical consequences. This study aims to evaluate the neurophysiological responses of police academy cadets during stressful decision-making scenarios and to predict individual stress levels from those responses. Fifty-eight police academy cadets from three cohorts watched a custom-made, naturalistic video scene and then chose the appropriate course of action. Simultaneous 32-channel electroencephalography (EEG) and electrocardiography (ECG) captured brain and heart activity. Event-related potentials (ERPs) and band-specific power features (particularly delta) were extracted, and machine-learning models were trained with nested cross-validation to predict perceived stress scores. Global and broadband EEG activity was suppressed during the video stimulus and did not return to baseline during the cooldown phase. Widespread ERPs and pronounced delta-band dynamics emerged during decision-making, correlating with both cohort rank and self-reported stress. Crucially, a combined EEG + cohort model predicted perceived stress with an out-of-fold *R*^2^ of 0.32, outperforming EEG-only (*R*^2^ = 0.23) and cohort-only (*R*^2^ = 0.17) models. To our knowledge, this is the first study to both characterize EEG dynamics during stressful naturalistic decision tasks and demonstrate their predictive utility. These findings lay the groundwork for neurofeedback-based training paradigms that help officers modulate stress responses and calibrate decision-making under pressure.

## 1. Introduction

Stress is an inherent component of the demanding roles fulfilled by police and military personnel [1,2,3]. They frequently operate in high-stakes environments, where rapid and effective decision-making is crucial. Understanding the underlying neural mechanisms of stress that influence decision-making in such contexts is essential for enhancing the performance, safety, and mental health of these professionals [4,5]. The identified neural mechanisms could serve as biomarkers to assess personnel readiness for specific tasks or could be incorporated into biofeedback systems to complement existing training regimens [6,7].

The prefrontal cortex is crucial for cognitive control and decision-making [8]. Additionally, prefrontal–limbic–striatal networks modulate and drive optimal decision-making and performance [9]. Neuroimaging studies have investigated the neurophysiological correlates of stress and their impact on memory retrieval, decision-making, and attention. The structure and functions of the prefrontal cortex can impair working memory retrieval even under mild uncontrollable stress [10]. Furthermore, stress can induce a shift in memory systems in both animals and humans, leading them to favor “habit” memory systems over cognitive ones, which are primarily modulated by the hippocampus and prefrontal cortex [11].

Several studies have investigated the electrophysiological and neural correlates of decision-making through electroencephalography (EEG) recordings and revealed the complex spatiotemporal dynamics of EEG signals associated with decision-making tasks. They have found that the alpha-band power consistently decreases under psychosocial stress, whereas the beta-band power tends to increase [12]. Successful decision-making is associated with theta band activity in the parietal and central regions during the process of memory recall in decision-making tasks [13]. The decision-making process also exhibits varying temporal dynamics that indicate different stages, such as task identification, decision-making, and outcome evaluation [14]. Moreover, urgency and interference accumulation exhibit distinct neural correlates during the decision-making process [15].

The stimuli used in typical EEG experimental paradigms often do not reflect real-life scenarios, as these experiments are usually conducted in a laboratory or office setting rather than in the field. For instance, standardized stimuli, such as images, sounds, or videos, are employed for analyzing the neural correlates of emotional responses. Although this approach can minimize variance across studies, it fails to represent real-life scenarios that the participants may encounter. Recently, there has been a shift toward using naturalistic stimuli to better mimic real-life situations and record neurophysiological activity in more naturalistic settings [16]. Virtual reality (VR) has been used to fully immerse participants in a particular visual stimulus, thereby enhancing their engagement in the task. Naturalistic stimuli have been shown to increase their attention and task engagement while having a minimal impact on EEG quality [17]. Therefore, this study employed custom-made videos recorded and edited by a local acting and production team as the stimuli to mimic the real-life scenarios faced by police officers. To the best of our knowledge, this is the first study to use tailored naturalistic videos in an EEG-based experimental setting. Furthermore, we have been able to leverage EEG activity to quantify specific spatiotemporal neural biomarkers that correlate and predict perceived stress scores in response to our custom-designed naturalistic stimuli. This study builds upon our prior work [18], which analyzed the same dataset to examine neurophysiological correlates of situational performance. In contrast, the present article focuses on EEG-based prediction of perceived stress responses during decision-making.

## 2. Methodology

### 2.1. Data Description

The participant recruitment, experimental paradigm, and EEG/ECG recording procedures have been described previously [18]. The participants were 58 males, aged 18–30 (mean age = 22.8, SD = 3.2), recruited from the Saad Al-Abdullah Police Academy in Kuwait. All participants provided informed consent, and the experiment was approved by the Ethics Review Committee (ERC) of Kuwait University (Ethics Review Code: KU-ENG-14-11-23, approved on 11 November 2023). As the participants were students enrolled in the academy, they had already been evaluated to have no history of neurological disorders, psychiatric disorders, or drug use. The Saad Al-Abdullah Police Academy has two main programs: a 4-year program for high-school graduates and a 6-month program for university graduates. The participants were divided into three categories: first-year high-school (*n* = 21), third-year high-school (*n* = 17), and university graduates (*n* = 20). At the time of the experiment, the first-year cadets had six months of training, the university graduates had one year of training, and the third-year cadets had three years of training. They were asked to sit in a comfortable chair facing a computer screen. A 32-channel EEG acquisition system (g.Nautilus Multi-Purpose; g.tec medical engineering GmbH, Schiedlberg, Austria), with 31 channels dedicated to EEG and one to electrocardiography (ECG), was used for the measurements. The reference and ground electrodes were attached to the left and right mastoids of the participants, respectively. The EEG electrodes were arranged in a standard 10–20 system while the Oz electrode was repurposed for ECG acquisition. This ECG electrode was placed on the left rib area. The police academy protocol mandates that all enrolled students must maintain a shaved head. Therefore, before wearing the EEG cap, the scalp and mastoids of each participant were cleaned with a disinfectant wipe. In this study, we used dry electrodes instead of wet ones. Although wet electrodes are the gold standard for EEG recording, dry electrodes also perform reasonably well [19]. The signal-to-noise ratio of the g.SAHARA electrodes is lower than that of other dry-wireless EEG systems [20]. Additionally, the ease of use and quick interchangeability of dry electrodes made them well-suited for this study because the experiment was conducted with approximately 20 cadets per day, owing to the limited time window available for experimentation.

A novel experimental paradigm was established to simulate a realistic and stressful decision-making scenario comprising local dialects, customs, and criminal activities. A local acting agency was hired to capture scenes simulating a domestic abuse call. The video was captured from the perspective of the police officer, who responded to the call by arriving at the scene. It comprised seven individual events, where the video was paused and a 5 s prompt was shown, asking the participant what their response would be at that specific moment. The prompt consisted of three Arabic words that translate to “What is your course of action?” and centered on the screen to minimize eye movements. They were asked to internalize their decisions to reduce movement artifacts by articulating their responses. Each response is of similar complexity, and the participants were given 5 s to make the decision. For example, in one particular event, a kidnapper brandished a knife and held the victim, threatening to kill them if the police officer did not put his gun down. The experiment was designed to elicit emotional reactions and decision-making processes. The video was displayed using Presentation (Neurobehavioral Systems, Inc., Albany, CA, USA), a stimulus delivery and experiment control program. The MMBT-S Trigger Interface Box (Neurospec AG, Stans, Switzerland) was used to precisely pinpoint the stimulus onset and synchronize it with the EEG recordings. The video stimulus was 4 min long. Before the stimulus, a 3 min baseline video was shown, depicting a person walking along a paved walkway on a beach in Kuwait. Additionally, a similar 3 min cooldown video was shown after the stimulus ended. This was done to establish the baseline neural activity for each cadet and to investigate the extent to which the cadets returned to baseline after engaging in a stressful emotional scenario. Both the cool down and baseline videos use similar beach-walking footage, matched for luminance, motion content, and audio levels, ensuring a valid return-to-rest comparison. Once the experiment was completed, each subject was asked to move to a separate room, where they were interviewed to rate their perceived stress scores using a Likert scale, ranging from 1 (lowest) to 5 (highest), for each of the seven critical scenarios. The lowest and highest possible total stress score for the entire stimulus period is 7 and 35, respectively. Figure 1 illustrates the experimental paradigm used in this study.

### 2.2. Data Preprocessing

EEG data were sampled at 500 Hz and band passed between 1 and 40 Hz using a Butterworth infinite impulse response (IIR) filter. The bandpass data were cleaned using the automated artifact rejection function of EEGLab. Bad channels were identified as those that were flat for more than 5 s and removed, and the minimum accepted correlation between the nearby channels was 0.8. The built-in artifact subspace reconstruction function of EEGLab for bad burst correction was used to correct the data with a maximum standard deviation of 20 for each 0.5 s window. Finally, if the maximum of 25% of channels exceeded 12 standard deviations above the mean for each 0.5 s window, the data of those segments were removed. Once the data were cleaned, independent component analysis was used to remove artifacts corresponding to non-brain-related activities, such as eye blinks and facial muscle movements. Independent components with a classification probability ≥90% for eye or muscle artifacts (as identified by EEGLAB’s ICLabel) were rejected and removed from the data. On average, 3.12 ± 1.66 channels (≈10% of electrodes) and 1.26 ± 0.71 independent components (≈4–5% of ICs) were rejected per recording. For event-related potential (ERP) analysis, epochs were extracted from the trigger markers at −2–5 s relative to the stimulus onset. Subsequently, the ERPs were normalized to neural activity from −2 s to stimulus onset. In addition to the EEG data, the ECG data were sampled at 500 Hz and synchronously collected. Heart rate variability is heavily modulated by the autonomic nervous system, which is segmented into the sympathetic and parasympathetic nervous systems. The sympathetic nervous system is responsible for the “fight or flight” response, and the parasympathetic nervous system is commonly referred to as modulating the “rest and digest” response. Stressful, emotional situations typically increase sympathetic nervous system activity, which can be assessed using numerous heartrate variability metrics. In this study, we used the normalized low-frequency activity of the heart-rate variability. To achieve this, the R-peaks in the ECG waveform were detected, and the RR interval was calculated by determining the time difference between adjacent R-peaks. However, the RR interval was not sampled uniformly; therefore, it was interpolated using cubic spline interpolation and sampled at 4 Hz. The RR interval was then detrended using the smoothness priors algorithm to remove ultra-low-frequency components (<0.035 Hz) that could distort the subsequent spectral analysis, which required stationary signals [21]. The powers of the two frequency bands were determined. First, the low-frequency band (0.04–0.15 Hz), which is related to both parasympathetic and sympathetic activation, and the high-frequency band (0.15–0.4 Hz), which is related to parasympathetic activation, were calculated. The high-frequency component is negatively correlated with acute mental stress [22]. The normalized high-frequency component of the spectral power (HFnorm) was calculated by dividing the high-frequency power by the total power. This metric has been previously employed to quantify the proportion of parasympathetic to sympathetic activity [23,24], as well as an index of parasympathetic nervous-system modulation [25].

### 2.3. Data Analysis

#### 2.3.1. Autonomic and Neural Average Activity

As mentioned in the previous section, the experiment consisted of three segments: baseline, stimulus, and cooldown. First, we investigated whether each segment of autonomic tone and neural activity exhibited a similar trend. The HFnorm was calculated for each participant for each segment. A reasonable assumption would be that the sympathetic activity of the participants was the highest when they were watching the 4 min stimulus video and lowest when they were watching the initial baseline video. An ANOVA test was used to determine whether differences existed in the means of the HFnorm values in each segment. This analysis was repeated independently for each group to identify group-dependent trends. The average neural activity during each segment was assessed by first applying principal component analysis (PCA) to the clean continuous EEG recordings pooled across all subjects within each cohort. The top ten spatial components (PCs) explaining the highest variance were retained. Each subject’s EEG was then projected onto these components to extract principal component scores. For each PC, power spectral density (PSD) estimates were computed using Welch’s method across three time segments: baseline, stimulus, and cooldown. As opposed to analyzing different frequency bands separately, investigating the PSD will determine whether periodic or aperiodic activity is changing during the experiment [26]. Broadband power was calculated by integrating the PSD from 1 to 30 Hz. To identify components that consistently captured neural dynamics across cohorts, only robust PCs—those with highly correlated spatial loadings (*r* > 0.8) across groups—were retained. Paired *t*-tests were conducted to compare broadband power during the stimulus and cooldown segments to baseline, and false discovery rate (FDR) correction [26] was applied to account for multiple comparisons.

#### 2.3.2. Event-Related Potential Analysis

Event-related potentials (ERPs) were extracted individually for each EEG channel. Each participant completed seven trials corresponding to distinct decision prompts, with each trial segmented from –2 to 5 s relative to stimulus onset. ERPs were computed separately for each group, based on 140 trials for university students, 147 trials for first-year high school students, and 119 trials for third-year high school students. Additionally, all trials across groups were pooled to generate overall ERPs representative of the entire sample. To investigate spatiotemporal neural dynamics, topographical plots of neural activity were generated across time. Spectral dynamics were assessed by filtering the ERPs into four canonical frequency bands using Butterworth IIR filters: delta (1–4 Hz), theta (4–8 Hz), alpha (8–12 Hz), and beta (12–32 Hz) [27,28]. The analytic signal was obtained via the Hilbert transform, and the resulting signal envelope was smoothed using a moving-average filter to mitigate edge-related oscillations. Frequency-specific ERPs were then derived to track temporal changes in band-limited activity. To examine the relationship between spectral activity and perceived stress, average spectral power was computed for each participant from 0 to 5 s after stimulus onset and then averaged across the seven trials, resulting in one feature vector per channel per participant. Perceived stress scores were summed across trials to produce a single outcome score per participant. For channels showing statistically significant correlations, linear regression models were fitted to quantify the relationship between spectral activity and perceived stress. Finally, for frequency bands that show significant correlations between stress and neural activity, frequency-specific ERPs where there are statistical differences in activity between cohorts were determined.

#### 2.3.3. Cross Validated Stress Prediction Pipeline

To assess the predictive utility of EEG activity for estimating subjective stress, we implemented a nested cross-validation (CV) framework using ridge regression. EEG features were extracted from a 3 s window (2–5 s post-stimulus onset) and averaged across time, trials, and channels for each participant, resulting in a feature vector representing mean activity per subject. This time window was selected based on a parameter search evaluating model performance across multiple possible windows, to avoid the effect of early evoked potentials, and to isolate later sustained neural activity. Data were collected from three cohorts—university students (*n* = 20), first-year high school students (*n* = 21), and third-year high school students (*n* = 17)—and merged to create a full dataset of 58 participants.

The perceived stress scores for each subject in each critical scenario were summed to yield a total stress score per subject. The stress scores were then log-transformed to reduce skewness and stabilize variance, thereby improving linear model assumptions and reducing the influence of outliers. Predictive modeling was performed using repeated 5-fold cross-validation, stratified by cohort. Across 20 random repetitions, three models were trained and compared: (1) a cohort-only model using dummy-encoded group membership as predictors, (2) an EEG-only model using the top *K* (*K* = 10) EEG features with the strongest absolute correlation to stress scores (selected on training data only), and (3) a combined model incorporating both EEG and cohort features. All features were standardized based on the training set.

Each model used ridge regression with a fixed regularization parameter (*λ* = 0.2). Ridge regression was chosen for its ability to handle multicollinearity and reduce overfitting in high-dimensional feature spaces, which are typical of EEG data. Model performance was quantified using out-of-fold predictions and computing the coefficient of determination (*R*^2^) for each repeat. The added predictive value of EEG was assessed by calculating ∆*R*^2^ between the combined and cohort-only models. Predicted stress scores were back-transformed to the original scale for visualization.

To evaluate generalizability and robustness, we reported the mean and standard deviation of *R*^2^ across all 20 repetitions for each model. Additional analyses included residual plots and histograms of mean absolute error (MAE). Collectively, this approach provided a comprehensive assessment of how well EEG features alone, and in combination with demographic information, could predict perceived stress levels.

## 3. Results

### 3.1. Overall Effects of Video Stimulus

The three phases of the experiment were labeled “Baseline,” “Stimulus,” and “Cooldown.” A one-way ANOVA test was applied to investigate whether there were any changes in the normalized high-frequency component (HFnorm) between the three phases of the experiment for each cohort group. No statistical differences were found, which indicates that any autonomic response could not have been captured in the ECG signal. Figure 2 displays principal components (PCs) that were robust across all three cohorts and showed statistical changes in the power spectrum during the three stages of the experiment. PCs were considered robust if their spatial topography (i.e., loading pattern) exhibited a Pearson correlation *>* 0.8 with at least one PC in each of the other cohorts. There is a broadband downwards shift in the PSD when comparing Stimulus and Cooldown to Baseline for all three cohorts. Most noticeably, the average power spectrum of the Cooldown phase did not return to Baseline but remained suppressed. Paired *t*-tests for this PC confirmed that this suppression is statistically significant after correcting for multiple comparisons (*p* < 0.05). For all three cohorts, this PC explained most of the variance in the EEG activity and is spatially global with a centroparietal emphasis. All of the PC loadings were positive, which emphasizes that it is indeed a global shift.

### 3.2. Event-Related Potentials Analysis

During the video stimulus, there were seven pauses during which the participants were asked to mentally determine the appropriate course of action based on the situation. ERPs during these events were analyzed to investigate whether there was a significant effect post-stimulus compared to pre-stimulus. Prior to statistical analysis, outlier trials were removed. The strongest post-stimulus ERP effects were observed over front-temporal channels as shown in Figure 3 where the top 4 channels with the highest Cohen’s d scores are shown. Notably, channel F7 showed the largest effect size with Cohen’s *d* = 0.66, based on *n* = 388 valid trials (*t*(387) = 13.09, *p* < 0.0001). Similarly, robust effects were found in T7 (*d* = 0.56, *t*(390) = 11.15), FC5 (*d* = 0.55, *t*(389) = 10.77), and F8 (*d* = 0.53, *t*(374) = 10.20), all with *p* < 0.0001. 9 out of 31 total channels have a Cohen’s d score of greater than 0.4 when comparing average post-stimulus activity to pre-stimulus. The shaded regions represent the standard error of the mean. In channel F8, immediately after the stimulus onset (vertical black line), we can observe a negative deflection around the 150 ms mark, a positive deflection around the 650 ms mark, and a final negative deflection around the 1 s mark. By contrast, nearly opposite results are evident in channel F7. Specifically, we can observe a very prominent positive deflection at the 350 ms mark, a negative deflection at the 650 ms mark, and a final positive deflection at the 1 s mark. In all exemplar channels shown, a very low-frequency oscillatory activity (approximately 1 Hz) is evident after the 1 s mark. In all exemplar channels shown, a very low-frequency oscillatory activity (approximately 1 Hz) is evident after the 1 s mark. Both channels T7 and FC5 mirror the behavior of channel F7, albeit with a weaker amplitude deflection.

Figure 4 shows the topological projections across the scalps of the participants from stimulus onset for up to 5 s. Primarily, three main regions were activated: the parietal-occipital lobe and the left and right frontal-temporal areas, which were mainly indexed by channels F7 and F8. First, 100–200 ms after stimulus onset, we can observe an increase in the activation of parietal-occipital channels and a decrease in the frontal region. Secondly, from 200 to 800 ms, we can observe the same phenomenon as that in Figure 3. Then, there is a positive deflection on the left and a negative deflection on the right, and then the polarity switches near the 600 ms mark. Subsequently, the polarity of the left and right frontal areas switches again at the 900 ms mark, and an additional increase in activity in the parietal-occipital area is evident. Finally, the ERP returns to baseline after 1200 ms.

As described in the Section 2, we investigated whether the frequency components of these ERPs could be correlated with stress. We applied a channel-wise Pearson correlation between the average post-stimulus activity of the four canonical frequency bands and overall stress scores. Only post-stimulus delta-band activity exhibits channel-wise correlations with stress scores. Figure 5 shows the channels that survived multiple comparison testing and that exhibited a statistically significant linear correlation between stress scores and the instantaneous amplitudes of the delta-band activity. Pearson correlation analysis revealed significant negative associations, indicating that increased EEG amplitude was consistently associated with lower perceived stress. Specifically, six channels (FP1, FP2, AF3, F8, FC5, and CP5) demonstrated significant negative correlations that survived multiple comparison correction (FDR-adjusted *p* < 0.05).

Figure 6 displays delta-band activity as a function of cohort rank. Statistically significant differences were present in two channels, AF3 and FC5. A one-way ANOVA was conducted to compare EEG amplitude differences across the three cohorts at each electrode. After applying false discovery rate (FDR) correction for multiple comparisons (*q* < 0.05), two channels showed significant group differences. Specifically, AF3 showed a significant effect of cohort on average EEG amplitude (*F* (2, 403) = 5.95, *p* = 0.0028, adjusted *p* = 0.044), as did FC5 (*F* (2, 403) = 7.97, *p* = 0.0004, adjusted *p* = 0.013). These findings indicate that frontal EEG activity varied systematically by cohort at these sites.

### 3.3. Stress Prediction

Figure 7 displays the results of the cross-validated stress prediction framework as described in the Section 2. The dashed black line corresponds to perfect prediction. Firstly, there is information contained in the cohort that relates to perceived stress scores since different training amounts would lead to group-level differences in stress response (*R*^2^ = 0.17). The EEG only, which excludes cohort identity, yields a higher predictive performance (*R*^2^ = 0.23). However, when including both cohort rank and post-stimulus delta EEG activity, a combined model is able to perform much better than a cohort only model, with a ∆*R*^2^ = 0.15, highlighting the utility of EEG in predicting stress responses in a timescale that is in the order of seconds. The ∆*R*^2^ was not sensitive to initial randomization in the cross-validation scheme, as 19 out of 20 iterations showed significant performance of the combined model compared to the cohort-only model (Appendix A). The MAE histogram and residuals of the combined model are also demonstrated in Appendix A. To confirm that our cross-validation approach is not sensitive to the chosen time window of 2 to 5 s post-stimulus onset, a hyperparameter search was conducted where the start time and end time of the time window were adjusted in 0.5 s increments. The performance of the classifier did not significantly change as a function of the time window, as long as it is chosen to capture temporal dynamics occurring in the later stages of the stimulus. The highest-performing time window was from 3 to 5 s (Appendix A). We also evaluated the stability of the predictive model across a range of ridge penalties (λ = 0–1 in steps of 0.1). Model performance remained essentially unchanged across this range (∆*R*^2^ < 0.01), confirming that the predictive results are robust to the choice of λ. For consistency, subsequent analyses were conducted with λ fixed at 0.2. Finally, to evaluate whether a nonlinear model could improve prediction performance, we trained a shallow ANN with one hidden layer (8 ReLU units) and a linear regression output under the same cross-validation and feature selection methodology as above. Combined *R*^2^ performed consistently below baseline (mean *R*^2^ = −0.52).

## 4. Electrooculography (EOG) Proxy Analysis

Since the bulk of our results are driven by delta band activity, one possible confound is that horizontal or vertical eye movements are contaminating our results. Firstly, to evaluate whether ocular activity contributed to the broadband suppression observed in our PCA results, we derived horizontal (HEOG) and vertical EOG (VEOG) proxies from the EEG channels. HEOG was computed as the difference between lateral frontal electrodes (F7–F8), capturing polarity shifts associated with horizontal saccades. VEOG was estimated as the average of the frontopolar electrodes (Fp1 and Fp2), which are maximally sensitive to blink-related vertical activity. Using these broadband (1–40 Hz) proxies, a two-way ANOVA on HEOG revealed a significant main effect of Phase (ie. whether in baseline, stimulus or cooldown) (*p* = 0.001) and Cohort (*p* = 0.029), but no interaction (*p* = 0.699). VEOG showed no effect of Phase (*p* = 0.425), a modest effect of Cohort (*p* = 0.011), and no interaction (*p* = 0.999). When pooling trials, neither HEOG (*p* = 0.172) nor VEOG (*p* = 0.284) differed significantly across groups.

Notably, HEOG activity decreased during stimulus presentation but rebounded in the cooldown. This is inconsistent with the sustained broadband suppression revealed by PCA. Furthermore, the PCA loadings were predominantly centro-parietal, whereas ocular artifacts are expected to present with a frontal distribution. These results support that our broadband PCA effects reflect neural activity rather than ocular contamination. In addition, we investigated whether HEOG/VEOG proxies are correlated with stress. At the subject level, neither proxy showed a reliable association with stress. HEOG exhibited a nonsignificant positive correlation (*r* = 0.16, *p* = 0.218), while VEOG was near zero (*r* = −0.05, *p* = 0.697), confirming that ocular activity was unrelated to the stress effects. Finally, we re-ran our stress prediction analysis after removing the frontal-pole channels (F8, F7, T7, T8) and observed no significant reduction in performance (combined *R*^2^ = 0.31).

## 5. Discussion

This study explored the neurophysiological underpinnings of decision-making in police cadets under stress using a naturalistic video stimulus designed to mimic a real-life scenario. The results provide new insights into the impact of stress on decision-making processes in high-stress situations and highlight the potential of EEG and ECG for assessing and training police officers. In contrast, heart rate variability measured from ECG did not change with stimulus onset or cool down, which highlights the value of integrating EEG in assessing stress and performance. In this study, we employed an unconventional approach to providing the stimulus, as it consisted of a custom-made video reflecting a local issue that police cadets could encounter in the field. The video was recorded in the local Kuwaiti dialect of Arabic. Compared with using standardized stress-inducing stimuli or videos that do not relate to regional issues, this approach is expected to keep police cadets engaged in the tasks at hand. Additionally, because of the nature of our stimulus, which was limited to a 4 min video, only seven decision-making scenarios or trials were included. However, even with the limited number of trials per participant, we were still able to extract the neurophysiologically relevant ERPs. Primarily, the results demonstrated that a more ecological approach to the experimental design can yield significant results.

### 5.1. Autonomic and Neural Effects of the Video Stimulus

There was no difference in autonomic tone, as indexed by the HFnorm metric, between the baseline, stimulus, and cooldown portions of the video across all three cohorts. The stimulus itself was insufficient to induce a detectable autonomic response in the ECG data, as also reported in our previous work [18]. However, from a neurological perspective, the EEG data showed that during the stimulus and cooldown periods, broadband activity decreased, with the most prominent decrease occurring in the centroparietal regions. This suppression of activity may reflect cognitive load and emotional stress experienced during decision-making tasks. During the cooldown phase, the broadband activity did not recover to baseline, which could indicate that the subjects were engaging in cognitive activity and evaluating their responses during that time. Both analyses demonstrate that when using a naturalistic video, EEG captures neurophysiological task-relevant changes more effectively than ECG. Consistent with our PCA findings of broadband spectral suppression during task engagement, several studies have demonstrated that cognitive load elicits widespread EEG desynchronization. Pei et al. [29] showed that increasing cognitive demand results in reductions in alpha and beta power across the scalp, a flattening of the 1/f spectral slope, and alterations in phase–amplitude coupling, indicating large-scale cortical reorganization. Similarly, Hanslmayr et al. [30] reported that successful memory encoding is accompanied by widespread decreases in alpha and beta power, suggesting that neural desynchronization supports richer information representation. The study by Lenartowicz et al. [31] further confirmed that high cognitive load results in spatially diffuse reductions in alpha power across multiple working memory tasks, indicating a task-general signature of desynchronization. These convergent findings support the interpretation that the first principal component in our data reflects a broadband, global reduction in spectral power—a robust neurophysiological marker of cognitive engagement.

### 5.2. Significant Event-Related Potentials (ERPs) During Decision Making

Even with a low number of trial counts per participant and the ecological nature of the video stimuli, we successfully extracted clear ERPs arising during the decision-making process. Figure 3 and Figure 4 show the asymmetrical negative and positive deflections in frontal channels F7 and F8, which primarily occurred at stimulus onset and lasted up to 1.5 s, which could indicate neural processes that are involved in emotional processing and decision-making [12,32,33,34]. The positive deflection in the parietal lobe at approximately 100 ms could be attributed to the visual processing of the stimulus. At approximately the two-second mark in Figure 3, we observed an increase in delta-band oscillatory activity. It has been shown that during concentration, delta band activity increases mainly in frontal EEG leads [35]. This increase in delta power is related to functional cortical deafferentation, which interferes with internal concentration [34]. Furthermore, the instantaneous amplitudes of the delta-band activities for two exemplar channels indicated that the delta-band activity increased in channel FC5, similar to the early ERP observed in Figure 3. However, it did not return to the baseline.

Similarly, the overall delta activity in AF3 increased at stimulus onset and remained elevated thereafter. This neural dynamic may be related to information processing and decision-making [14,15,36]. Furthermore, the differences in delta-band activity between cohorts, as shown in Figure 6, suggest that training and experience modulate delta-band activity. More experienced cadets, such as university graduates, exhibited higher peak delta activity than third- and first-year high-school graduates. Delta band activity of university graduates remained higher than that of the other cohorts throughout the stimulus onset. In FC5, the differences between the three cohort ranks are clear, with university graduates having a higher delta peak and a higher sustained delta, followed by third-year high school and first-year high school graduates.

### 5.3. Correlation with Perceived Stress and Prediction

A statistically significant correlation existed between the delta-band activities in channels Fp1, Fp2, Af3, F8, Fc5, and Cp5 and the perceived stress, suggesting that this can serve as a neural biomarker for assessing the stress responses of police officers when they are engaged in stressful scenarios. Moreover, this linear relationship suggests that as perceived stress increases, delta-band activity decreases, indicating that higher stress levels are associated with reduced neural efficiency in processing stressful stimuli. This finding aligns with those of previous research that found that stress can impair cognitive functions and decision-making. Additionally, EEG offers a finer temporal resolution than other biosignals, such as heart-rate variability, which must be measured at a temporal resolution on the order of minutes. Therefore, EEG can be an effective method for assessing the stress levels of individuals engaged in fast-paced tasks. Although we found a correlation between stress and the EEG results in this study, further studies are required to assess the possibility of determining the type of stress response. Many studies have demonstrated the potential of EEG for detecting emotion types [37]; therefore, a natural extension of our experiment would involve using different kinds of ecological stimuli to assess whether EEG can be used to detect emotional or cognitive stress.

After confirming that perceived stress scores are correlated with post-stimulus delta band activity, it is essential to determine whether this neural signal could be used as a biomarker for prediction. Our cross-validated approach, as shown in Figure 6, confirms that post-stimulus delta band activity, particularly in the later stages where information processing and decision-making are most likely to occur, can be used to predict perceived stress. In addition, our combined-only model can explain around 32% of the variance in stress scores. The information contained in the delta band activity provides insights beyond what cohort rank can offer, highlighting the usefulness and utility of using EEG for evaluating police cadets during stressful tasks. Importantly, the temporal precision of relevant EEG activity for stress prediction in our work is on the order of seconds, compared to heart-rate variability metrics, which typically require minutes.

### 5.4. Implications and Future Directions

In addition to the stress-response assessment, the results of this study can be used to develop a neurofeedback system to enhance police officers’ responses to stressful and emotionally charged scenarios. Previous studies have employed neurofeedback systems to enhance cognitive performance and mitigate stress [38,39,40,41]. The system would involve observing the delta-band activity during stressful tasks; hence, the system would require the individual to maintain a high level of delta-band activity, which was shown to correlate negatively with perceived stress as we found that this correlation is evident in the frontal channels, where relatively inexpensive EEG headbands can be used to assess stress responses, and personalized thresholds provide visual or haptic feedback. The average delta-band activity is relatively straightforward to extract using a causal signal processing pipeline; this data can be obtained and processed in real time.

In this study, we demonstrated the ability to extract meaningful ERPs and showed that delta-band features could be used to predict perceived stress. However, a significant limitation of this work is our method of collecting perceived stress scores, which was conducted through post-experimental surveys. As a next step, creating a real-time assessment of stress scores could improve the accuracy of our predictors in the model. For example, a prompt could be shown post-stimulus that asks the cadet to input their perceived stress using a keyboard or other input tools. In addition, using a prompt for answering questions could pinpoint precisely when the decision was made, which could lead to a richer and more sophisticated analysis of the data. Using a real-time prompt answering paradigm could provide additional metrics such as time-to-decision, which could index the cadet’s confidence in their answer. Another limiting factor of this work is the low number of trial counts per subject, seven. This is because we are using a custom-made video, which, by nature, would be very cost-prohibitive to create multiple scenes to increase the number of trials. Despite the small number of trials per subject, we were able to extract meaningful neural signatures from the data and use them to predict stress.

A natural extension of this work is to apply more sophisticated analyses, such as connectivity measures, to determine whether they relate to stress and cohort rank. It has been shown that resting-state connectivity is a predictor of neurofeedback performance [42]. More sophisticated multiscale features can also be investigated to determine if they improve decoding performance [43,44]. Deep learning and artificial intelligence are quite effective in decoding emotions, motor control, and speech from EEG [37,45,46] as they learn nonlinear representations of the data. Therefore, a deep-learning architecture can decode perceived stress based on neural activity data. In our results, a shallow ANN could not predict stress better than baseline, which is most likely due to our small sample size. However, by employing more training data, a neural network can be trained and used for stress assessment during police cadet training through additional experiments. Previous studies have implemented VR to fully address emergency police officers’ stress and firearm training [7,47]. Additionally, expanding the study to include female cadets and a broader age range could provide a more comprehensive understanding of the neural dynamics of stress and decision-making across diverse populations.

## 6. Conclusions

This study offered novel insights into the neurophysiological mechanisms underlying decision-making in police cadets under stress. A naturalistic video stimulus revealed significant changes in neural activity, highlighting the prolonged impact of stress on the brain. The correlation between delta-band activity and perceived stress underscored the potential of EEG metrics as biomarkers for stress assessment and training. Delta-band activity was used to predict perceived stress scores, highlighting their utility. These findings pave the way for future research and practical applications aimed at improving police officers’ resilience to stress and their decision-making capabilities.

## Figures and Tables

**Figure 1 sensors-25-05925-f001:**
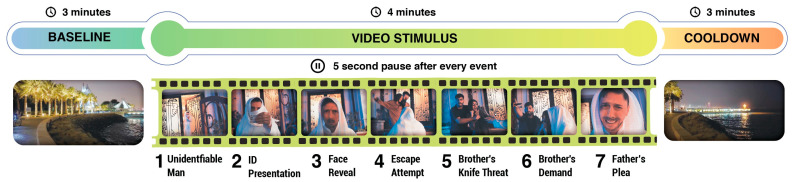
**Experiment flow.** The experiment comprised three videos: 3 min baseline, 4 min video stimulus, and 3 min cooldown. The entire video stimulus is a continuously shot scene that depicts a possible real-life scenario that could occur in the field. At seven different events during the experiment, a 5 s pause was employed wherein the participants were shown a prompt asking them to decide on the best course of action. Each event represents a unique scenario.

**Figure 2 sensors-25-05925-f002:**
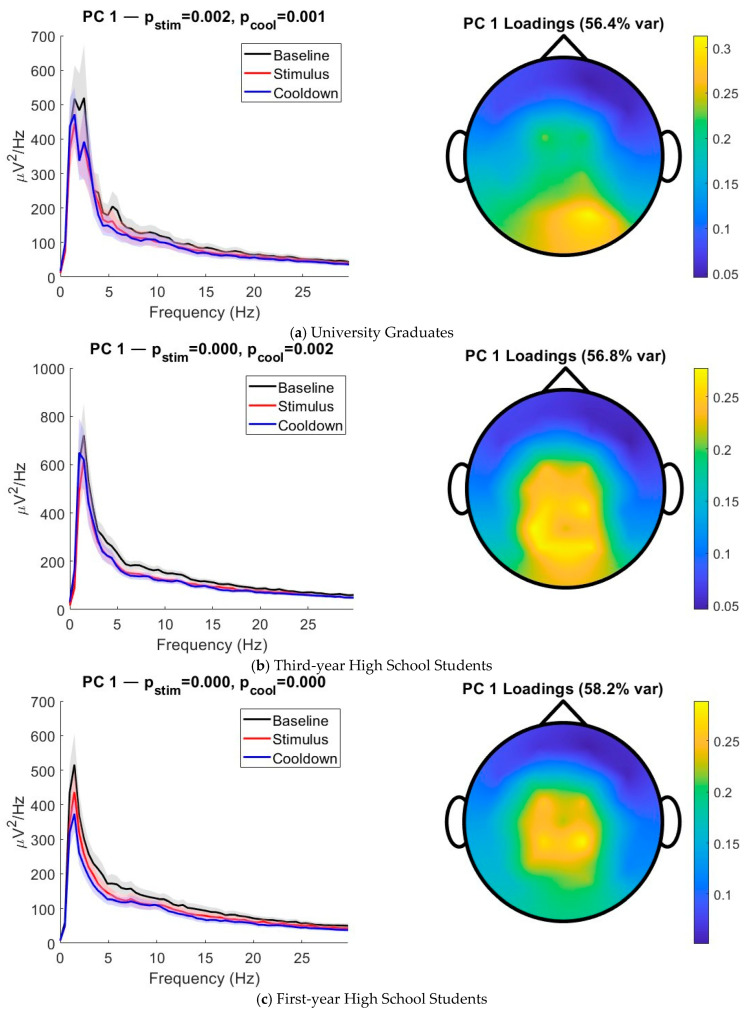
**Average activity of different frequency bands.** Spectral power and spatial topography of the first principal component (PC1) across cohorts. For each group—(**a**) university students, (**b**) third-year high school students, and (**c**) first-year high school students—the left panel shows the power spectral density (PSD) of PC1 during baseline (black), stimulus (red), and cooldown (blue) periods. Shaded regions indicate the standard error of the mean. Statistical significance of broadband power suppression during the stimulus and cooldown periods was assessed using paired *t*-tests relative to baseline, with resulting *p*-values reported above each plot. The right panel shows the scalp topography of the PC1 spatial loadings, with the percentage of variance explained indicated in the title. Across all cohorts, PC1 exhibited significant broadband suppression during the stimulus and/or cooldown periods and reflected a spatially global mode of neural activity, with peak contributions centered over midline and central scalp regions.

**Figure 3 sensors-25-05925-f003:**
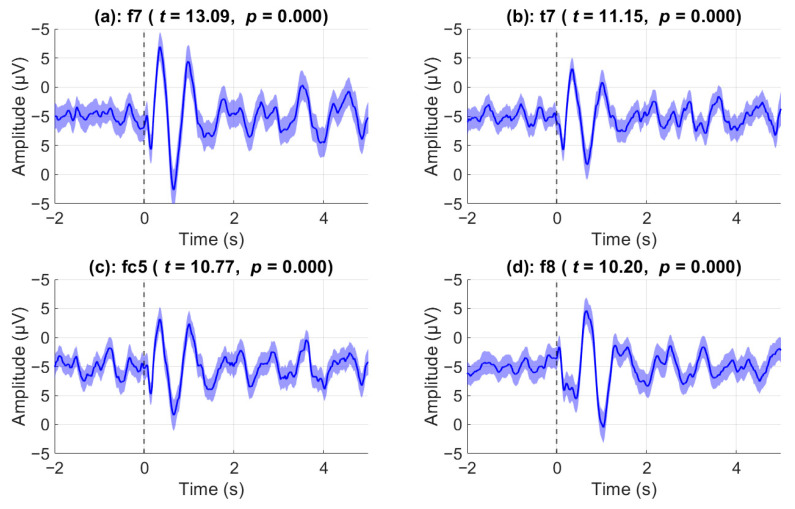
**Event-related potentials in top four significant channels.** Grand-average event-related potentials (ERPs) at the four channels showing the strongest post-stimulus responses. Channels were selected based on the largest effect sizes from a two-tailed paired *t*-test comparing post-stimulus (0–5 s) to pre-stimulus (–2 to 0 s) activity across all trials and participants. The plots display mean voltage (black trace) ± standard error (blue shading). All four channels (F7, T7, FC5, F8) showed statistically significant ERP responses (*p* < 0.001), with peak amplitudes occurring within the first second after stimulus onset.

**Figure 4 sensors-25-05925-f004:**
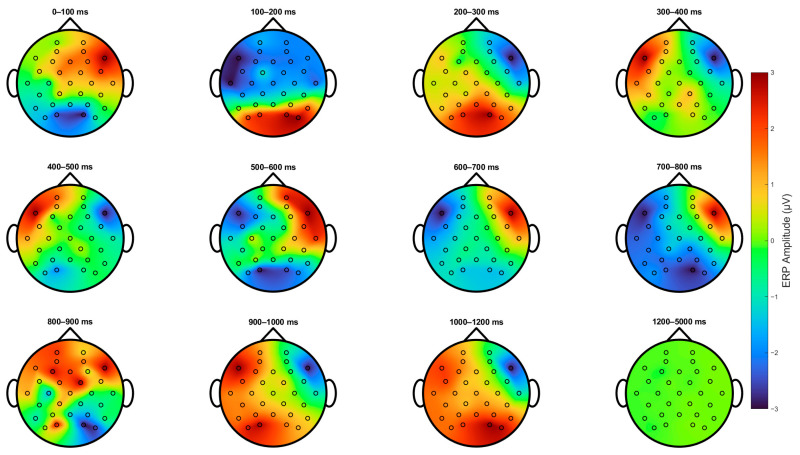
**Topological projection on the scalp of ERPs for all subjects.** Scalp topographies of event-related potential (ERP) amplitudes across twelve consecutive time windows following stimulus onset. Each map represents the mean ERP amplitude within a 100 ms interval, from 0–100 ms to 1200 ms, with an additional window capturing late activity from 1200 to 5000 ms. Positive amplitudes are shown in red, negative in blue, and neutral in green, as indicated by the color bar. The sequence captures the evolving spatiotemporal dynamics of the ERP response. Notably, between 200 and 800 ms, a clear asymmetrical activation pattern emerges over frontal regions, with stronger activity in the left hemisphere relative to the right. Activity diminishes after approximately 1200 ms, returning toward baseline.

**Figure 5 sensors-25-05925-f005:**
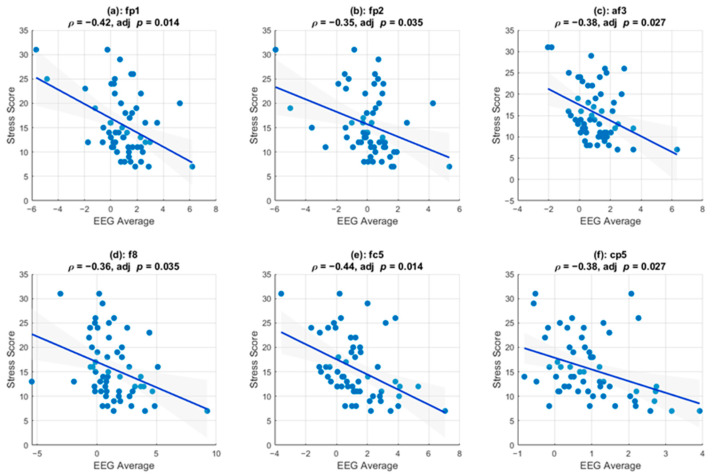
**Channel-specific associations between delta-band EEG (0–5 s Post-Stimulus) and stress score.** For six electrodes (Fp1, Fp2, Af3, F8, Fc5, Cp5), each scatter shows the participant’s average delta-band amplitude in the 0–5 s window after stimulus onset (x-axis) versus their stress score (y-axis). Blue lines are linear regressions with 95% confidence intervals shaded in gray. Above each panel are Pearson’s *ρ* and FDR-adjusted *p*-values; all six channels exhibit significant negative correlations (adjusted *p* < 0.05), indicating that higher post-stimulus delta activity is correlated with lower reported stress.

**Figure 6 sensors-25-05925-f006:**
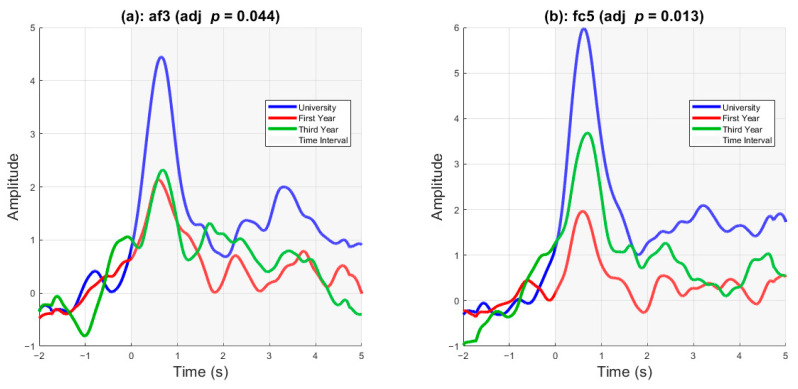
**Delta-band activity as a function of cohort rank.** Delta-band activity over time for channels AF3 and FC5, shown separately for university (blue), third-year high school (green), and first-year high school (red) cohorts. Delta-band activity was estimated by extracting the analytic amplitude from the Hilbert transform of delta-filtered (1–4 Hz) ERP signals, followed by smoothing with a moving-average filter. These exemplar channels were selected based on statistically significant differences in post-stimulus delta activity across cohorts (adjusted *p* < 0.05), using a one-way ANOVA followed by post hoc testing. Both channels exhibit a clear increase in delta activity following stimulus onset, with group-level differences emerging near the 1 s mark. In both AF3 and FC5, university students show the largest delta response, while first-year students show the weakest, reflecting a graded pattern of activation aligned with cohort rank. This pattern remains especially elevated in the university group beyond 3 s post-stimulus.

**Figure 7 sensors-25-05925-f007:**
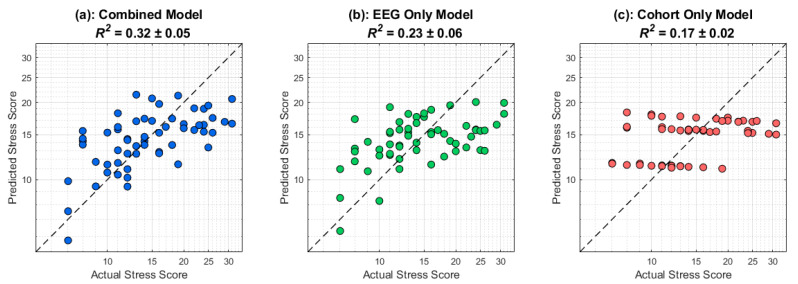
**Out-of-fold stress predictions demonstrate added value of EEG beyond cohort rank**. Scatterplots compare each participant’s actual self-reported stress (x-axis) against predicted stress (y-axis) under three feature-set configurations: (left) combined EEG and cohort demographics (*R*^2^ = 0.33), (center) EEG features only (*R*^2^ = 0.24), and (right) cohort demographics only (*R*^2^ = 0.16). Every point is from a held-out fold; the dashed diagonal is perfect prediction (y = x), and the solid line is the best-fit regression. Note that adding EEG activity to cohort rank yields a meaningful increase in explained variance over cohort alone, underscoring the unique predictive contribution of EEG features.

## Data Availability

Requests for access to the study data should be directed to the corresponding author. Sharing data is subject to the approval of Kuwait’s Saad AlAbdulla Police Academy.

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
