# Peer review of "EEG-Based Prediction of Stress Responses to Naturalistic Decision-Making Stimuli in Police Cadets"

_sensors, 2025, doi:10.3390/s25185925_

Round 1
Reviewer 1 Report
Comments and Suggestions for Authors
The study aims to find a correlation between the EEG signal patterns and the subjects’ stress levels, based on EEG recordings and feedback collected from police cadets of different ages. In my opinion, the data collection method is professionally sound, and the number of participants involved is sufficient for drawing generalized conclusions. However, the study has several weaknesses.
First, the authors did not make the collected data publicly available, which raises concerns about the reliability of the results. These should be made accessible. In addition, I believe that the linear model is too simplistic to capture the patterns in the data and results in a high error rate. I recommend using a regression-based artificial neural network as a secondary model.
Moreover, the authors relied on the frequency range between 1–40 Hz during the analysis. They need to justify this choice. In doing so, I would ask them to address the role of the frequently cited alpha, beta, etc. frequency bands. I would also like to recommend that the authors take the following two articles into consideration when addressing this point:
- Zhang, R. Fruchter, M. Frank, Are they paying attention? A model based method to identify individuals’ mental states, Computer, vol. 50, pp. 40-49, 2017.
- Suto, S. Oniga, Music stimuli recognition in electroencephalogram signal, Elektronika Ir Elektrotechnika, vol. 24, pp. 68-71, 2018.
Finally, the paper contains several formal issues and it does not follow well the manuscript template of the journal.
Reviewer 2 Report
Comments and Suggestions for Authors
This manuscript presents a novel investigation into EEG-based neural correlates of stress responses during naturalistic decision-making scenarios among police cadets. The authors have developed a commendable experimental paradigm, leveraging tailored naturalistic video stimuli that simulate high-stress situations pertinent to police duties.
Major:
- The predictive pipeline selects the top-K EEG features (K=10) by correlation on the training set and fixes the ridge penalty. While feature selection is kept within folds, the fixed λ may not be optimal and could bias comparative R².
- The Discussion positions ECG as potentially useful, yet the manuscript’s central positive findings are EEG-driven while ECG effects are not emphasized.
- Frontotemporal effects can be contaminated by eye movements and facial EMG—particularly in emotionally charged video viewing. Provide explicit preprocessing details (ICA/ASR criteria, channels/components rejected) and residual EOG proxies per condition; consider adding control analyses (e.g., re-running key statistics after removing frontal poles; comparing delta associations when excluding epochs with saccade-like transients). (Head shaving and dry electrodes are advantages for contact, but do not eliminate ocular/EMG artifacts.)
- Although the sample size (N=58) is adequate, the limited number of trials (7 per participant) could potentially affect ERP signal stability.
- In this or future work, authors are suggested to conduct multiscale EEG decoding analysis to enhance the interpretation of cognitive load and stress-related neural signatures (e.g., methods mentioned in "Understanding the role of eye movement pattern and consistency during face recognition through EEG decoding" and "EEG-based Familiar and Unfamiliar Face Classification Using Filter-Bank Differential Entropy Features".
Round 2
Reviewer 1 Report
Comments and Suggestions for Authors
The paper can be accepted.
Reviewer 2 Report
Comments and Suggestions for Authors
Authors addressed all my concerns. Thank you.